# Chromosomal scale assembly of parasitic wasp genome reveals symbiotic virus colonization

Jérémy Gauthier[1,2], Hélène Boulain[1,3], Joke J. F. A. van Vugt[4], Lyam Baudry[5,6], Emma Persyn[7], Jean-Marc Aury [8], Benjamin Noel[8], Anthony Bretaudeau[9,10], Fabrice Legeai[9,10], Sven Warris[11], Mohamed A. Chebbi[1], Géraldine Dubreuil[1], Bernard Duvic[12], Natacha Kremer [13], Philippe Gayral[1], Karine Musset[1], Thibaut Josse[1], Diane Bigot[1], Christophe Bressac[1], Sébastien Moreau[1], Georges Periquet[1], Myriam Harry[14], Nicolas Montagné[7], Isabelle Boulogne[7], Mahnaz Sabeti-Azad[7], Martine Maïbèche[7], Thomas Chertemps[7], Frédérique Hilliou[15], David Siaussat[7], Joëlle Amselem[16], Isabelle Luyten[16], Claire Capdevielle-Dulac[14], Karine Labadie[8], Bruna Laís Merlin[17], Valérie Barbe[8], Jetske G. de Boer[4,18,19], Martial Marbouty[5], Fernando Luis Cônsoli [17], Stéphane Dupas[14], Aurélie Hua-Van [14], Gaelle Le Goff[15], Annie Bézier[1], Emmanuelle Jacquin-Joly [7], James B. Whitfield[20], Louise E. M. Vet[4,18], Hans M. Smid[18], Laure Kaiser[14], Romain Koszul[5], Elisabeth Huguet [1], Elisabeth A. Herniou[1] & Jean-Michel Drezen[1✉]

Endogenous viruses form an important proportion of eukaryote genomes and a source of novel functions. How large DNA viruses integrated into a genome evolve when they confer a benefit to their host, however, remains unknown. Bracoviruses are essential for the parasitism success of parasitoid wasps, into whose genomes they integrated ~103 million years ago. Here we show, from the assembly of a parasitoid wasp genome at a chromosomal scale, that bracovirus genes colonized all ten chromosomes of *Cotesia congregata*. Most form clusters of genes involved in particle production or parasitism success. Genomic comparison with another wasp, *Microplitis demolitor*, revealed that these clusters were already established ~53 mya and thus belong to remarkably stable genomic structures, the architectures of which are evolutionary constrained. Transcriptomic analyses highlight temporal synchronization of viral gene expression without resulting in immune gene induction, suggesting that no conflicts remain between ancient symbiotic partners when benefits to them converge.

A list of author affiliations appears at the end of the paper.

Cotesia wasps (Hymenoptera, Braconidae) are parasitoids of Lepidoptera widely used as biological control agents to control insect pests[1,2]. Female wasps lay their eggs into caterpillars and larvae develop feeding on the host hemolymph. Parasitoid wasps evolved several strategies that increase parasitic success, including a sensitive olfactory apparatus to locate their hosts[3,4] and detoxification mechanisms against plant toxic compounds accumulating in their host (Fig. 1). However, the most original strategy is the domestication of a bracovirus (BV) shared by over 46,000 braconid wasp species[5]. Bracoviruses originate from a single integration event ~103 million years ago (mya) of a nudivirus (virus having a large DNA genome closely related to well-known baculoviruses) in the genome of the last common ancestor of this group[6–10]. Virus domestication confers a benefit to the wasps that use BVs as virulence gene delivery systems[5]. Indeed, virulence genes are introduced with wasp eggs into their hosts, causing inhibition of host immune defenses[5,11,12].

To gain insights into the evolution of endogenous viral sequences in the wasp genomes, we obtained a reference genome for Cotesia congregata at a chromosomal scale. Whereas endogenous viruses most often slowly decay after integration[13], we show that viral sequences colonized all the chromosomes, reaching a ~2.5-fold higher number of genes than a pathogenic nudivirus. However, the bracovirus is only partially dispersed across the wasp genome since specialized regions of up to two megabases are devoted to the production of packaged DNA and of viral structural proteins. Comparison with genome scaffolds of another wasp revealed a striking stability of these regions over 53 million years[14], suggesting strong evolutionary constraints. Expression patterns and molecular evolution of virus genes point to a central role of the viral RNA polymerase in maintaining the bracovirus as a domesticated but still identifiable viral entity. Despite massive virus particle production, wasp immune genes are not induced, suggesting no conflicts remain between the wasp and the virus.

## Results

**Genome assembly, annotation, and comparison**. We used a hybrid sequencing approach combining 454 reads, Illumina short reads, and chromosomal contact data (HiC), to obtain a reference genome for Cotesia congregata at a chromosomal scale. First, a 207 Mb high-quality genome (contig N50 = 48.6 kb, scaffolds N50 = 1.1 Mb and N90 = 65 kb) was obtained for C. congregata using a combination of mate pair 454 pyrosequencing and Illumina sequencing (Supplementary Data 1). Most of the assembly (86%) consisted of 285 scaffolds of over 100 kb. This genome was then reassembled based on experimentally obtained chromosomal contact maps. The HiC method yielded ten main scaffolds comprising >99% of the previously obtained genome assembly (Supplementary Data 1 and Supplementary Fig. 1), and corresponding to the ten chromosomes of C. congregata[15] (Supplementary Fig. 1). In addition, draft genomes of five related Cotesia species—C. rubecula, C. glomerata, C. vestalis, C. flavipes, and C. sesamiae—were sequenced and assembled with Illumina shotgun sequencing reads (Supplementary Data 1) for molecular evolution analyses on homologous genes. They respectively resulted in contig N50 values of 13, 9, 15, 20, and 26 kb and cumulative sizes of 216, 243, 176, 155, and 166 Mb.

The genome of C. congregata comprises 48.7% of repeated DNA sequences including 34.7% of known transposable elements (TEs) (Supplementary Fig. 2). Bracovirus proviral segments included TE sequences that had previously been annotated as bracovirus genes: we revealed that the BV26 gene family corresponded to miniature inverted-repeat transposable elements (MITE) derived from Sola2 elements abundant in the wasp

genome (Supplementary Fig. 2). This indicates that, contrary to a common paradigm[16], the virulence genes packaged in bracovirus particles do not exclusively originate from the wasp cellular gene repertoire.

We automatically annotated 14,140 genes in the genome of C. congregata (Methods and Supplementary Data 1), which include >99% of 1658 conserved insect genes (98 to 99% of the genes for the other Cotesia species, Supplementary Data 1). Then wasp genes potentially involved in the success of the endoparasitoid lifestyle, such as genes implicated in olfaction, detoxification and immunity, were individually annotated. This analysis performed on complex genes belonging to well-known families further assessed the quality of the genome obtained.

**Olfaction: highly dynamic evolution of olfactory receptors**. Manual annotation of chemoreceptor gene families identified 243 odorant receptor (OR), 54 gustatory receptor (GR), and 105 ionotropic receptor (IR) genes in C. congregata. These numbers are in the upper range of those of other parasitoid wasps, only slightly lower than in ants (Supplementary Data 2), whose large repertoires are attributed to the exploitation of complex ecological niches[17]. Phylogenetic analyses showed C. congregata ORs belong to 15 of the 18 OR lineages (Fig. 2) described in Apocrita[18] and revealed independent OR gene expansions in N. vitripennis and in the Braconidae (Fig. 2). The most spectacular Braconidae-specific expansions occurred in five clades each harboring at least 25 genes in C. congregata (Fig. 2 and Supplementary Data 2). Highly duplicated OR genes were found in 6 clusters of at least 10, and up to 19, tandemly arrayed genes (Supplementary Fig. 3). Within Braconidae, many duplications occurred in the ancestors of Cotesia, but OR copy numbers varied significantly between species (Fig. 2). This illustrates the highly dynamic evolution of OR gene families within parasitoid wasps and between Cotesia species, which have different host ranges. Although the link between genes and adaptation might be complex, this dynamic might be related to host search through the recognition of different volatile compound from insects and plants.

**Detoxification genes: a full set but no particular extension**. Genes from all families involved in detoxification in insects were identified by manual annotation in C. congregata, and are largely conserved within Cotesia (Supplementary Data 2). For instance, each species harbors conserved numbers of UDP-glucosyltransferases (UGTs) and slightly different numbers of gluthatione-S-transferases (GSTs). In contrast, carboxylesterases (CCEs) and cytochrome P450 (P450s) numbers vary widely with C. flavipes and C. sesamiae harboring few representatives (respectively, 22–24 CCEs and 49 P450s), compared to the 32 CCEs of C. rubecula and the 70 P450s found in C. congregata, which are both exposed to plant toxic compounds (Supplementary Data 2). Cotesia-specific P450 families were identified in the clan 3 and 4, both of which are often associated to adaptation to plant compounds and insecticides[19] (Supplementary Data 2). Altogether, Cotesia appear fully equipped for detoxification; however, in contrast to the OR genes, no spectacular gene expansion was observed. This suggests exposure to plant toxic products could be lower than expected on this third trophic level.

**Extension of bracovirus in wasp genome**. Imbedded in wasp DNA, the virus genomes have been extensively rearranged[20] since nudivirus integration. BV sequences (Fig. 1) are differentiated as (i) genes involved in particle production and named "nudiviral" genes (based on clear phylogenetic relationship within the Nudiviridae) and (ii) proviral segments packaged as dsDNA

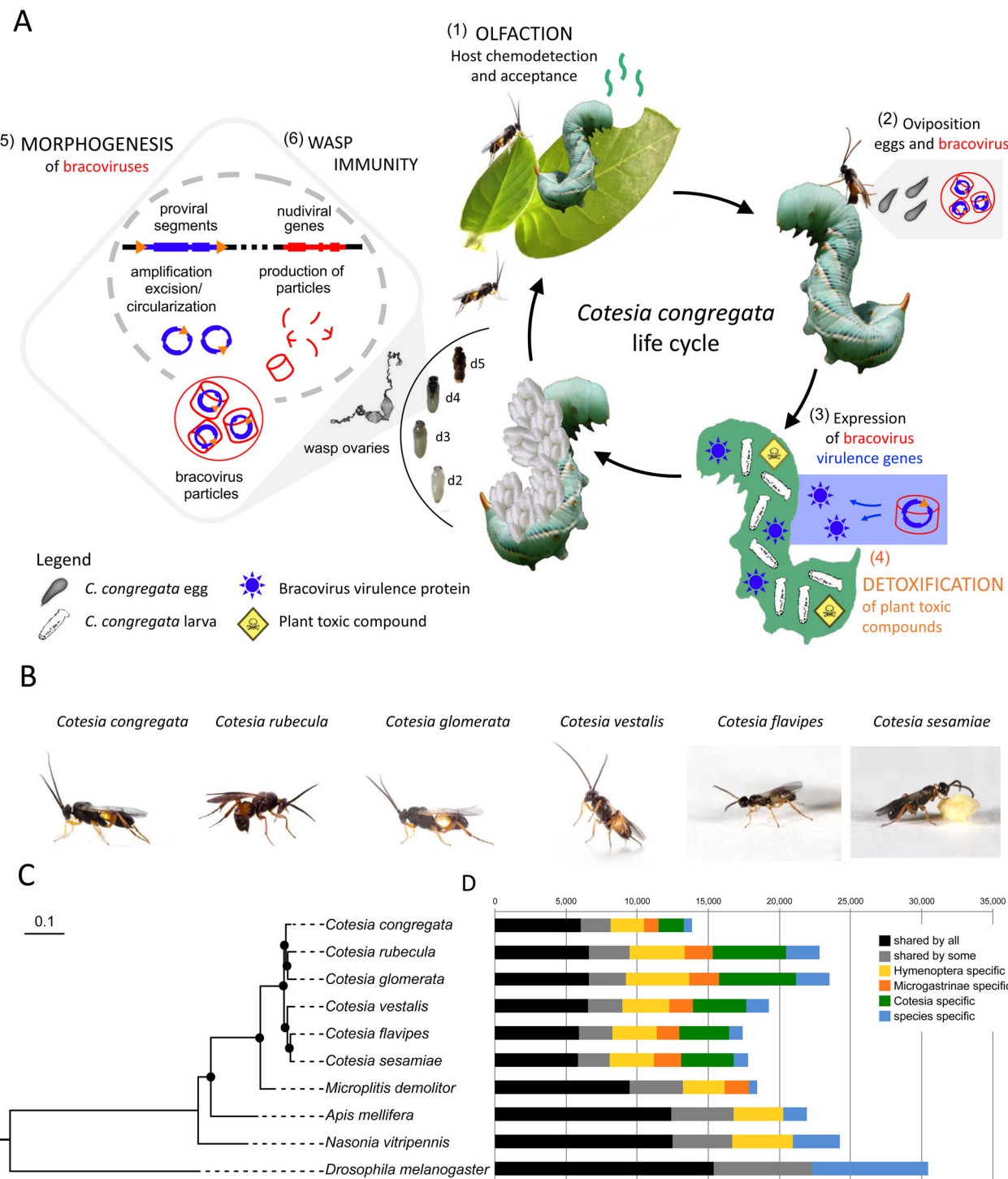

**Fig. 1 Cotesia species life cycle, phylogeny, and gene content. A** Major traits involved in the parasitoid koinobiont lifestyle and genome content of six *Cotesia* species. First (1) OLFACTION plays an important role in the detection of the plant (tobacco) attacked by caterpillars and host (*M. sexta*) larvae acceptance by adult wasps (*C. congregata*). Once the host is accepted, the wasp injects its eggs bathed in ovarian fluid filled with bracovirus particles (2). Bracovirus particles infect host cells, from which expression of bracovirus virulence genes (3) alter host immune defenses, allowing wasp larvae development (the eggs laid in the host body would otherwise be engulfed in a cellular sheath of hemocytes). As the host ingests plant toxic compounds, such as nicotine, while feeding, wasp larvae consuming the hemolymph containing these compounds rely on (4) detoxification to complete their life cycle. However, in these species associated with endogenous viruses the most important trait for parasitism success consists in (5) bracovirus morphogenesis during wasp metamorphosis, using genes originating from a nudivirus ancestrally integrated in the wasp genome. As massive production of virus particles occurs within wasp ovaries, (6) wasp immunity may be induced during particles production; d2, d3, d4, d5 refer to developmental stages of *C. congregata* larvae[36]. **B** Pictures of the six *Cotesia* species sequenced (credit H. M. Smid and R. Copeland). **C** Phylogeny of these species based on 1058 single-copy orthologous insect genes including the Microgastrinae *Microplitis demolitor* and outgroups (*N. vitripennis*, *A. mellifera*, and *D. melanogaster*). Black dots highlight branches with at least 90% support from maximum-likelihood analysis (1000 bootstraps). **D** Distribution of shared genes at several phylogenetic levels. Full protein-coding gene sets were included to identify orthologous gene groups. The "shared by some" category refers to genes shared by at least nine species among the ten studied. Note that the lower number of genes for *C. congregata* probably reflects the higher quality of the genome assembly obtained.

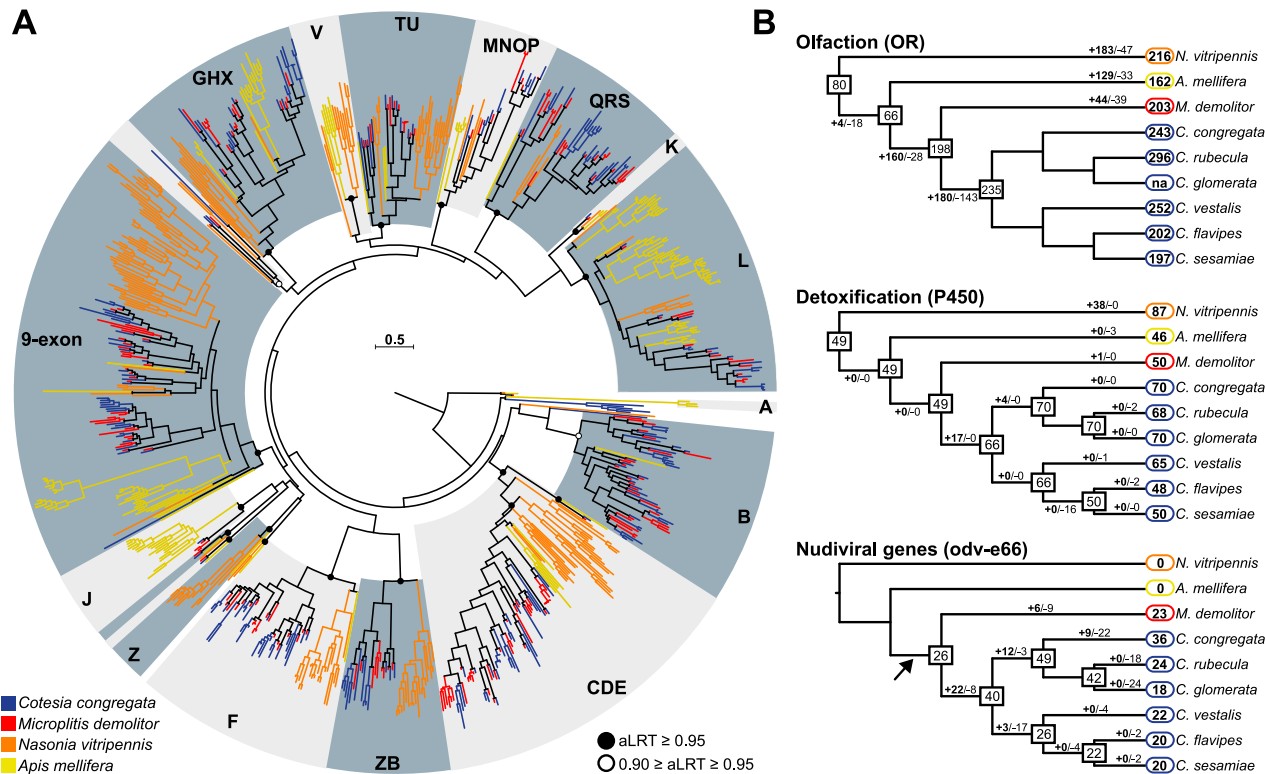

**Fig. 2 Gene family extensions in *Cotesia*. A** Maximum-likelihood phylogeny of the OR family in *C. congregata* and four other Hymenoptera species. The dataset included 243 amino acid sequences from *C. congregata* (blue), 203 sequences from *M. demolitor* (red), 216 sequences from *N. vitripennis* (orange), 162 sequences from *A. mellifera* (yellow). The tree was rooted using the Orco (OR-coreceptor) clade. Circles indicate nodes strongly supported by the approximate likelihood-ratio test (black circles aLRT ≥ 0.95; white circles 0.90 ≥ aLRT ≤ 0.95). The scale bar represents 0.5 expected amino acid substitutions per site. ORs of the five Hymenoptera species are distributed into 18 OR subfamilies previously described in[18] delineated in gray. **B** Copy number dynamics of OR (olfaction) P450 (detoxification) and *Odv-e66* genes, note that the later are found specifically in bracovirus-associated wasps since they derive from the ancestrally integrated nudivirus. Estimated numbers of gene gain and loss events are shown on each branch of the species tree. The size of OR repertoires in common ancestors is indicated in the boxes. The lack of phylogenetic resolution for closely related Cotesia OR genes precluded any comprehensive analysis of gene gains and losses.

circles in viral particles[20], encoding virulence genes which are involved in successful parasitism, and are similar to insect genes[21] or specific to bracoviruses[20].

The complete genome annotation of *C. congregata* revealed 102 nudiviral gene copies that have colonized all ten chromosomes. This number is similar to that of pathogenic nudiviruses[22], an unexpected result given that endogenous viral elements usually undergo gene loss in the course of evolution, with the exception of genes conferring protection against infections from related viruses[13]. Here, this surprisingly high number of nudiviral genes results from the balance between gene losses and the expansions of certain gene families. At least 25 of the 32 nudivirus core genes involved in essential viral functions[23] have been retained in the wasp genome, with the notable absence of the nudiviral DNA polymerase (Supplementary Fig. 5). The *fen* genes, generally involved in DNA replication, form a gene family with six tandem copies that is found specifically in the *Cotesia* lineage (Fig. 3B). The most spectacular expansion, comparable to those of OR genes, concerns the *odv-e66* gene family, which is typically found in one or two copies in nudivirus genomes[22], but is present as 36 genes in 10 locations (Figs. 2B and 3C), including 6 clusters of 2 to 10 copies, in *C. congregata*. This expansion occurred both before and after the divergence between *C. congregata* and *M. demolitor*[24,25], since we found tandemly duplicated copies in homologous loci of both species or in *C. congregata* only (Fig. 3D). In baculoviruses, *odv-e66* encodes a viral

chondroitinase[26] involved in digesting the peritrophic membrane lining the gut, thus allowing access to target cells during primary infection. We hypothesize that different ODV-E66 proteins may similarly allow BVs to cross various host barriers, and BV infection to spread to virtually all Lepidoptera tissues[27,28], thus differing from baculoviruses, whose primary infection is restricted to the gut and rely on a particular virion phenotype ("budded virus") to spread within their host. The large and continuous *odv-e66* gene family expansion we unravel here has most likely played an important role in wasp adaptation. One might speculate that during host shifts of the wasp, bracovirus particles might encounter different barriers, which would require adaptation by competitive evolution of duplicated *odv-e66* gene copies in a gene for gene coevolutionary framework[29].

**Genomic architecture and synteny of bracovirus genes.** Chromosome scale genome assembly of *C. congregata* provides for the first time the comprehensive genomic organization of a bracovirus within the genome of a wasp, allowing us to assess whether nudiviral genes[24] and proviral loci[20,21] that were previously found in different genome scaffolds could nevertheless be localized in the same chromosomal region. Examination of the very precise map of viral sequences within chromosomes (Fig. 3A, C, E) reveals a complex picture, since bracovirus sequences (nudiviral plus virulence genes) are indeed dispersed and present in all the chromosomes; however, the vast majority of them are

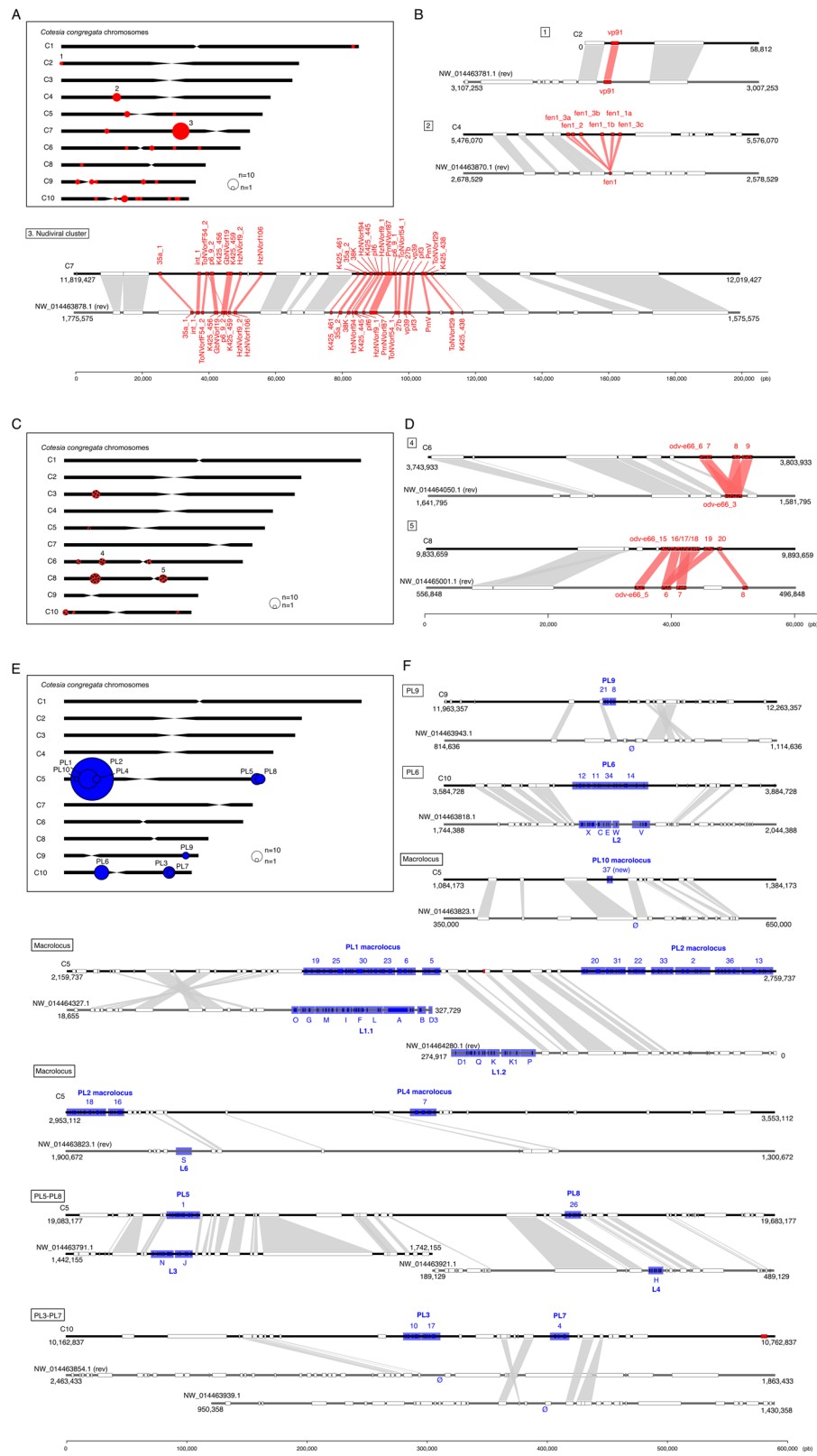

organized in clusters. Half of the single-copy nudiviral genes are located in the ~100 Kb nudiviral cluster, which comprises 25 genes (Supplementary Fig. 7). Comparison with the scaffold of *M. demolitor* showed almost perfect gene content and gene order conservation, as well as conserved syntenic blocks in the genomic regions flanking nudiviral sequences, over ~53 million years of evolution (Fig. 3B). This confirms that the nudiviral clusters of

both species are orthologous and likely derive from a genome fragment of the nudivirus that originally integrated in the ancestor wasp genome. This striking stability suggests that major evolutionary constraints maintain these genes together. The other nudiviral genes are dispersed in the wasp genome, although not evenly, as many more loci are located in the smallest chromosomes (Fig. 3A, C) and only one locus in the 4 largest

**Fig. 3 Synteny of nudiviral genes loci and proviral loci (PL) between *C. congregata* and *M. demolitor*. A** *C. congregata* chromosome map with the position of 24 nudiviral genes loci. **B** Comparisons between nudiviral gene regions of *C. congregata* and *M. demolitor*. Synteny between the two species has been characterized by at least two hymenopteran (non-viral) orthologous genes in the vicinity of homologous nudiviral gene(s) of both species. Genome scaffolds are represented in black. Red boxes indicate nudiviral genes and white boxes refer to hymenopteran genes. 1. the *vp91* region is orthologous indicating the position of this gene was inherited from their common ancestor 53 mya; 2. the *fen* region is also orthologous but an expansion occurred specifically in *Cotesia* lineage giving rise to six copies; 3. the organization of the nudiviral cluster encoding in particular capsid genes has remained strikingly similar with the same viral genes in the same order (except *p6-9-2*) in both species indicating strong evolutionary constraints. **C** *C. congregata* chromosome map with the position of gene loci corresponding to the highly expanded *odv-e66* nudiviral gene family. **D** Comparison of two *odv-e66* loci showing that expansion occurred before (cluster 7) and after (cluster 4) the separation of both species **E** *C. congregata* chromosome map with the position of Proviral Loci (PL) encoding virulence genes packaged in bracovirus particles. Note the concentration of loci (successively PL10-PL1-PL2 and PL4) in a 2 Mb region termed "macrolocus" and representing half of the chromosome 5 short arm. **F** Comparison of *C. congregata* and *M. demolitor* PL. Numbers 1 to 37 and letters correspond to the different dsDNA circles present in CcBV and MdBV particles produced from the PL. Blue boxes indicate virulence genes while white boxes refer to hymenopteran genes and the red boxes to a nudiviral *odv-e66* gene located between PL1 and PL2 and *58b* near PL3-PL7. Ø indicates the absence of orthologs PL in the *M. demolitor* genome.

chromosomes. Orthology with *M. demolitor* could be identified for 20 nudiviral gene regions (Fig. 3D and Supplementary Fig. 7), indicating they were already dispersed in the last common ancestor of both wasps and have stayed in the same loci. Altogether, this showed that nudivirus gene loss and dispersion occurred during the early period of wasp-bracovirus association (100 to 53 mya).

The expansion of virulence genes is another aspect of wasp genome colonization. In *C. congregata*, the 249 virulence genes encoded by proviral segments are concentrated in 5 regions of the genome located on three chromosomes; indeed among the eight proviral loci previously described[20], several were found localized in the same chromosomal region (PL5-PL8, PL3-PL7, PL2-PL4, Fig. 3E). Moreover, 77% of these genes clustered in a single region, which comprises four physically linked proviral loci (PL10-PL1-PL2-PL4) interspersed by wasp genes (Fig. 3E). This major virulence gene coding region (~2 Mb, 177 genes, 17 segments), which we named "macrolocus", is impressive since it spans half of chromosome 5 short arm and can be compared in size and gene number to the Major Histocompatibility Complex (MHC, ~4 Mb, ~260 genes)[30], which plays a major role in mammalian immunity. Orthology relationships was inferred between the PL1 in *C. congregata* macrolocus and the largest proviral region (comprising 11 segments) of *M. demolitor*[24] (Fig. 3E, F) but the macrolocus has undergone larger expansion in the *Cotesia* lineage (producing PL2, PL4, and PL10). Further syntenies were found between 5 isolated proviral loci (Fig. 3F), showing they were also already present in the common ancestor of *Cotesia* and *Microplitis* lineages 53 million years ago, and indicating that the global organization of the viral genome in wasp DNA was set up earlier. Overall most of the proviral loci are ancient, except the three localized loci in the long arm of chromosome C9 and C10 (PL3 and PL7, PL9 comprising 20 genes), the sole genuine novelties in the *Cotesia* lineage that appeared within the last 53 million years (Fig. 3F).

**Strong conservation of the bracoviral machinery.** The DNA circles packaged in bracovirus particles are produced following the genomic amplification of replication units (RU) that span several proviral segments of PLs[31,32]. Our detailed genomic analyses of *C. congregata* data led to the identification of a specific sequence motif at each RU extremity (Fig. 4D and Supplementary Fig. 4B) for both previously described types of amplification, associated with either head-tail or tail-tail/head-head concatemers[33] (Fig. 4D and Supplementary Fig. 4B), whereas a motif was previously identified for only one type[33]. We also confirmed the presence of circularization motifs[20,21] on all proviral segments at the origin of packaged circles (Fig. 4D and Supplementary Fig. 4B), indicating the conservation of a single viral mechanism

whatever the localization of viral sequences (Fig. 4 and Supplementary Fig. 4).

The conservation of viral functions in wasps over 100 million years of evolution is outstanding. Synonymous to non synonymous substitution ratio analyses on orthologous nudiviral genes in *Cotesia* showed most nudiviral genes (65 genes among the 79 tested genes) are evolving under stabilizing selection that is, however, less stringent than on the set of conserved insect genes used to assess genome completeness (Fig. 4C and Supplementary Fig. 5). This selection is notably strong for genes involved in viral transcription ($dN/dS < 0.08$), such as the RNA polymerase subunits (*lef-4, lef-8, lef-9, p47*), which most likely control nudiviral genes expression and, consequently, bracovirus particle production[6,10]. In contrast, genes involved in infectivity (homologs of baculovirus *pif* genes) appear less conserved (Supplementary Fig. 5). This might reflect divergence occurring during host shifts, through adaptation of virus envelope proteins to particular host cell receptors. The large *odv-e66* gene family and duplicated genes (*p6.9_2, pif-5_2, 17a*) similarly displayed less stringent to relaxed selection (Fig. 4 and Supplementary Fig. 5), which might be conducive to mutational space exploration for adaptation by neo-functionalization or sub-functionalization[34]. Virulence genes encoded by proviral segments globally displayed low conservation (Fig. 4), as expected for genes interacting with host defenses and involved in evolutionary arms race or adaptation to new hosts[35].

**Synchronized nudiviral transcription precedes bracovirus production.** The onset of bracovirus particle production has been detected using molecular biology and transmission electronic microscopy late during metamorphosis, 4 days after larvae have emerged from the caterpillar[36]. Previous experiments studying a handful of nudiviral genes during *C. congregata* pupal development showed a strong calyx specificity and unexpectedly early expression of a gene involved in nudiviral transcription[6,10]. We used RNAseq analysis to assess, for the first time, the expression of the complete set of 102 nudivirus genes in the ovaries throughout pupal development. The aim was to investigate in detail viral gene expression timing and whether nudiviral genes were synchronized. Genes involved in nudiviral transcription are highly expressed at day 2, but unexpectedly a large set of nudiviral genes transcripts is also detected at that time (Fig. 5C). Altogether, these results suggest that the onset of nudiviral gene transcription very quickly follows the production of the viral RNA polymerase, as would typically occur within 12 h in baculovirus infection. Afterwards at day 3, nudivirus gene expression has reached a much higher level, which could reflect that more cells are undergoing virus replication. The level then increased more slowly and reached a maximum at day 5 (Fig. 5, Supplementary Data 4, and Supplementary Fig. 6), when virus particle

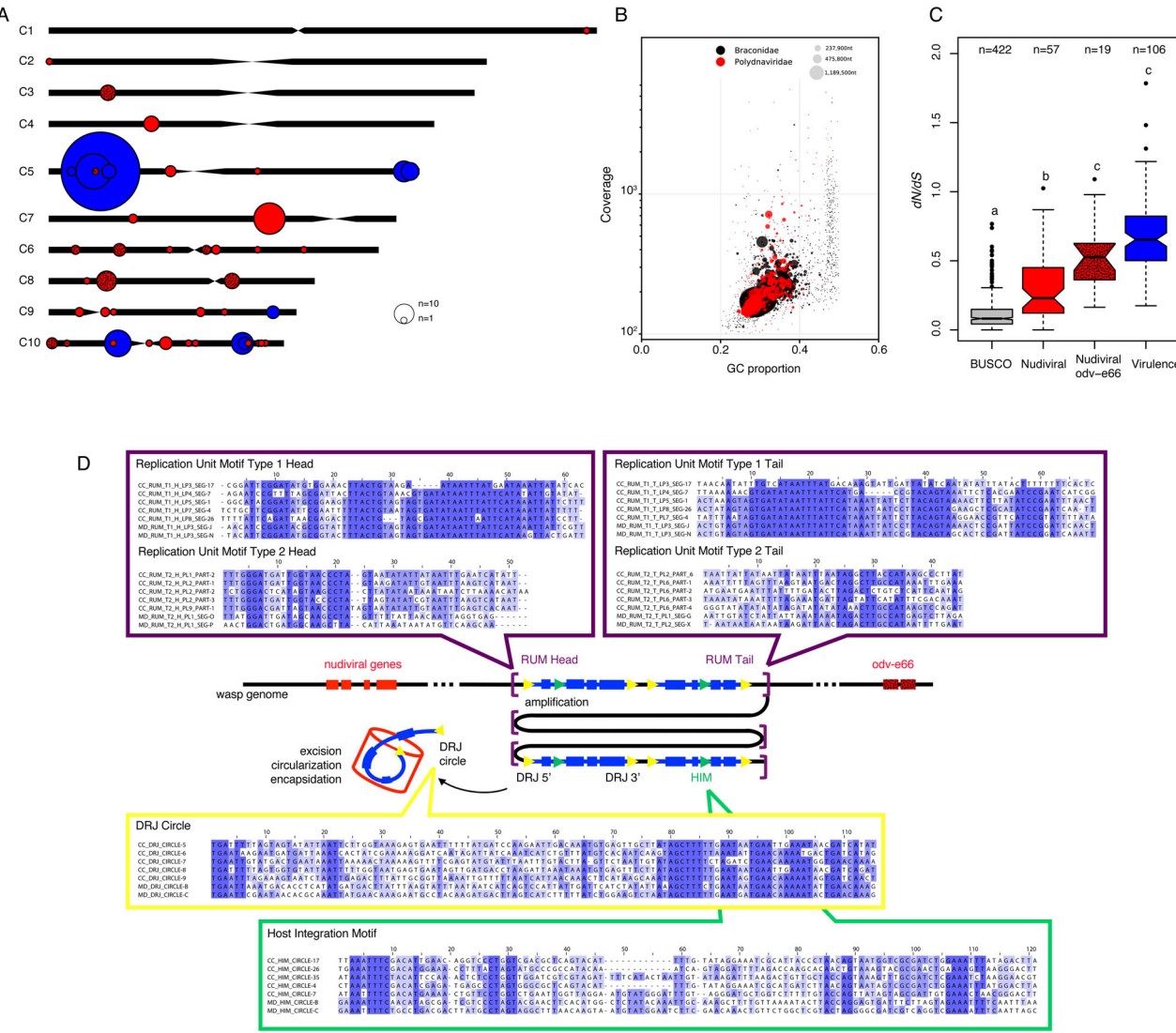

**Fig. 4 Bracovirus genes and motifs architecture and evolution. A** *C. congregata* chromosome map with the location of all bracovirus loci: nudiviral gene loci are shown in red, nudiviral *odv-e66* gene loci in hatched red and Proviral Loci (PL) in blue. The sizes of the circles correspond to the relative number of genes in each locus **B** Taxon-annotated GC content-coverage plot of the *C. congregata* genome associated with Braconidae (in black) and Polydnaviridae (in red). Each circle represents a scaffold in the assembly, scaled by length, and colored by taxonomy assigned by BlobTools. The *x*-axis corresponds to the average GC content of each scaffold and the *y*-axis corresponds to the average coverage based on alignment of the Illumina reads. **C** Measure of selection pressure on hymenopteran conserved genes, nudiviral genes and virulence genes. Pairwise evolutionary rates (dN/dS) of single-copy orthologous BUSCO genes, nudiviral genes, different copies from the expanded *odv-e66* nudiviral gene family and virulence genes of *C. congregata* and *C. sesamiae*. Letters above boxes indicate significant differences determined by Kruskal–Wallis test (H = 296.8, 2 d.f., *P* < 0.001) followed by post hoc comparisons. **D** Schematic representation of the genomic amplification during the production of viral particles in the wasp ovaries. Replication Unit Motifs (RUM) are the motifs that constitute the extremities of the molecules amplified during particle production. Direct repeat junctions (DRJ), at the extremities of each segment are used during the excision/circularization process to produce packaged dsDNA circles from the amplified molecules. Host integration motifs (HIM) are motifs used during the integration of bracovirus circles in host genome. For each of these motifs an alignment of a representative set of sequence comprising five motifs from *C. congregata* and *M. demolitor* are represented (complete alignments are shown in Supplementary Fig. 4).

production is the highest. Genes involved in viral transcription displayed a different pattern, since they already reached high level expression at day 2 and decreased significantly during virus production either from day 4 or day 5 (Fig. 5B). This time shift between expression of transcription and other nudiviral genes supports the hypothesis that the nudiviral RNA polymerase controls the expression of the other viral genes. The gap between nudivirus gene expression (day 2) and the onset of previously determined particle production (day 4) could reflect that very few cells may initially be involved in replication, and thus that high-throughput sequencing is required for their early detection. This hypothesis has important implications for studying how virus production is initially and selectively induced in the ovaries, which remains a major knowledge gap for understanding the wasp-bracovirus relationship.

Although nudiviral gene expression is variable, many transcripts reached impressive levels, similar to what would be expected of regular viral genes during infection. Indeed 12 nudiviral genes are among the top 50 of most expressed genes in *C. congregata* ovaries at day 5 (Supplementary Data 4). Moreover, three genes from the nudiviral cluster (including the major capsid component vp39) are by far the 3 most expressed of all wasp

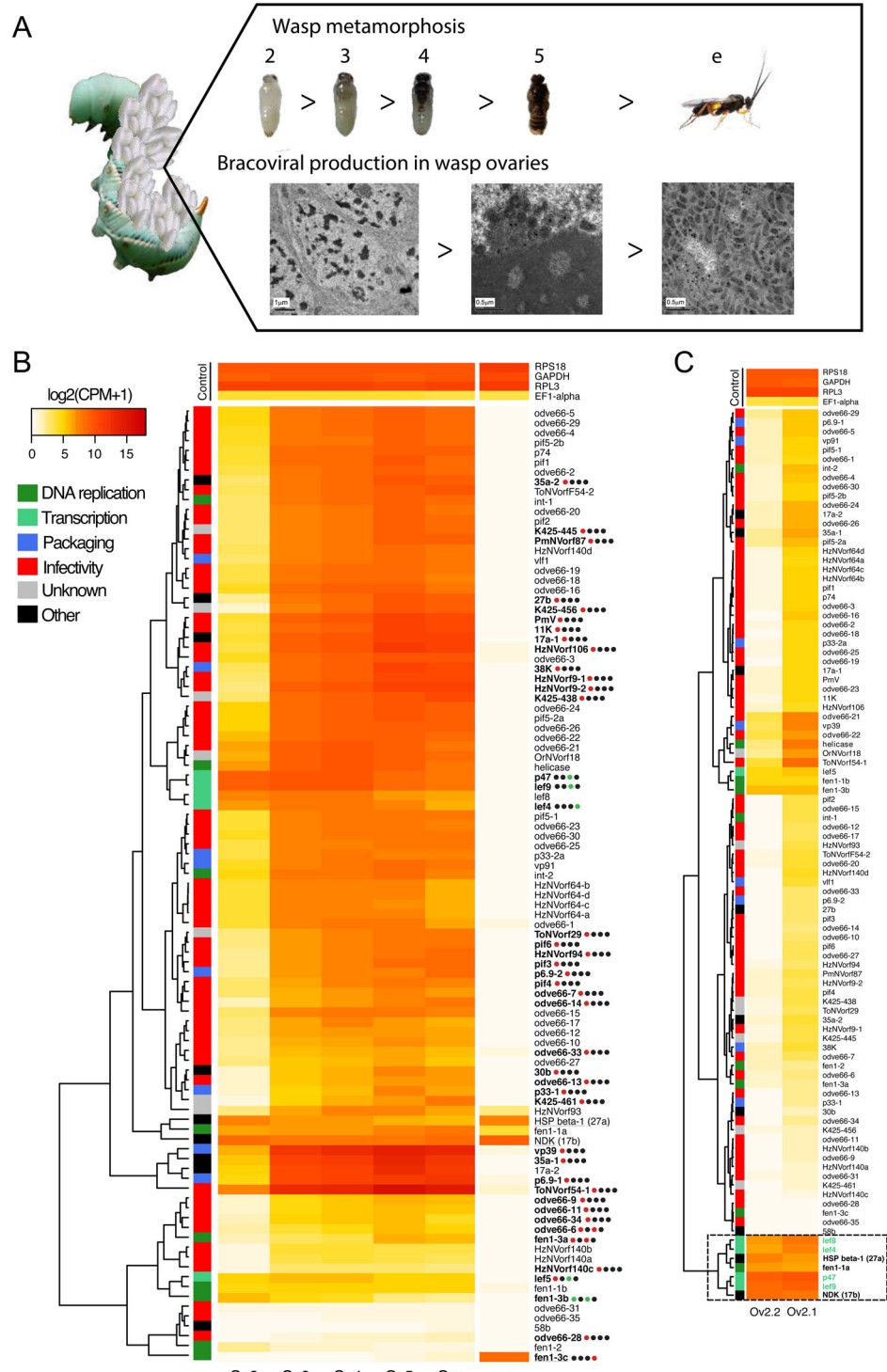

genes, suggesting virus particle production might mobilize most replicating cells transcription activity.

The analysis of venom gland transcripts, however, revealed some exceptions to nudiviral gene tissue-specificity, since 9 out of the 102 studied nudiviral genes are expressed in the venom gland (Fig. 5B). The fen-3c gene, for example, reached a high level in the venom gland and showed no expression in the ovaries. Moreover, transcripts of some of the gene copies belonging to extended gene families are barely detected in any samples (odve66-31, odve66-35) (Fig. 5B); they may correspond to pseudogenes, or, like fen-3c, be expressed in other tissues, for

a new function, no longer related to bracovirus production. For the vast majority of nudiviral genes, however, expression remains strongly synchronized during pupal ovarian development, turning on at day 2 and being already high at day 3 (Fig. 5B). Considering the age of the nudivirus wasp association, this reinforces the idea of strong evolutionary constraints.

**Immune gene expression during bracovirus production.** After 100 million years of endogenous evolution within the wasp genome, one can question whether virus particles produced massively in the ovaries are considered as a pathogenic virus by

**Fig. 5 Gene expression of nudiviral genes in the ovaries during *C. congregata* nymphal development. A** Pictures of wasp developmental stages studied and characteristic electron micrographs of ovarian cells involved in particle production. From d2 to d4 stage, cells that will produce particles show enlarged nuclei with chromatin condensation (left panel). From d5 massive particle production begins, particle assembly occurs at the periphery of a zone of electron dense material in the nucleus named "virogenic stroma" (middle panel). In newly emerged wasp nuclei are completely filled with bracovirus particles (right panel). Credit: Juline Herbinière. **B** Unsupervised hierarchical clustering based on gene expression in ovaries and venom glands. Ov2, Ov3, Ov4, Ov5 refer to the different ovaries across wasp developmental stages. Ove and vg refer to ovaries and venom glands of adult wasps. The colored squares associated with the clustering tree indicate the viral functions to which different nudiviral genes are supposed to contribute based on those of their baculovirus homologs. Heatmap of expression levels of 95 nudiviral genes is shown in the middle panel. Bold names highlight the genes that are validated as significantly differentially expressed between two consecutive stages using the statistical analysis and dots represent the four different comparisons studied between ovary stages (Ov2 vs. Ov3, Ov3 vs. Ov4, Ov4 vs. Ov5 and Ov5 vs. Ove). Black, red, and green dots indicate similar, increased and reduced expressions between consecutive developmental stages, respectively. The increase of some nudivirus genes expression between d2 and d3 visualized on the heat map was not validated statistically for all of them because in one of the d2 duplicates (shown in **C**) nudiviral genes expression had already reached high levels. Underlined genes show higher expression in venom glands compared to ovaries (Ove) note that 27a and 17b are not nudiviral genes but wasp genes, the products of which have been identified in Chelonus inanitus bracovirus particles. The expression of four stable wasp genes having high (RPS18, RPL3, GAPDH) or low expression level (EF1-alpha) is presented as control for comparison. **C** Unsupervised hierarchical clustering of gene expression from the two replicates of Ov2 ovary stage, that are very different regarding nudiviral gene expression levels, although dissected nymphae presented a similar coloration pattern, the left one representing a slightly earlier stage from the analysis of the whole set of wasp genes. Note that the genes within the box are already expressed at a high level in the earlier stage, including all of the genes involved in nudiviral transcription (shown in green) except lef5, in accordance with the hypothesis that the nudiviral RNA polymerase complex controls the expression of the other genes (lef-5 is associated with the complex but not a part of it).

the wasp, in which case their production should trigger an immune response. Whether viral production interacts in any way with the wasp immune system has remained totally unknown, however. Globally, the annotation of immune-related genes indicated *C. congregata* has an arsenal of 258 immune genes that are potentially involved in recognition, signal transduction, different signaling pathways, melanization and effector functions (Supplementary Data 2), in accordance with the recently reported annotation of *C. vestalis* immune genes[37]. We identified all members of the Toll, IMD, Jak/STAT and RNA interference pathways found in Hymenoptera (Supplementary Data 2).

In contrast to the sharp increase in nudiviral gene expression, no significant changes in immune gene expression could be detected in the ovaries during pupal development (Fig. 6, Supplementary Data 4, and Supplementary Fig. 6). In particular, expression of the genes involved in antiviral immunity (encoding members of the RNA interference, Jak/STAT or Imd/JNK pathways) was high in ovaries, even at stages before particle production is observable (ovaries stages 2 to 4), but hardly fluctuated during the course of ovary development, even at day 5, when massive particle production becomes apparent by TEM (Fig. 5A). Thus, no immune response appears to be induced or repressed at the cellular level as a response to high level nudiviral gene expression and particle production.

## Discussion

To investigate the genome features related to the endoparasitoid lifestyle of species associated with endogenous bracoviruses, we sequenced the genomes of six *Cotesia* species, and obtained an assembly at the chromosomal scale for *C. congregata*. This approach provided insights into genes potentially involved in essential functions of the wasps, such as olfaction, detoxification and immunity, as well as into the genomic evolution of the bracovirus. Large OR gene diversifications are often associated with host localization and acceptance. Indeed, female wasps rely on a sensitive olfactory apparatus to locate their hosts from volatile cues plants emit in response to herbivore attacks[3], and to assess caterpillar quality before oviposition[38]. Interestingly, OR copy numbers varied significantly during the evolution of the genus *Cotesia* (Fig. 2). The high dynamics of OR repertoire might point to the need for more specific recognition of chemical cues from the host and its food plant. Characterization of OR gene

sequences is the first step toward determining their function, and experimental settings using *Drosophila* cells are available for the identification of the volatile recognized by each receptor. This is of particular interest for future research on the modification of host acceptance through genome editing, to improve parasitoid strains used in biological control.

In contrast, no comparable diversification was observed in the detoxification arsenal, even though *Cotesia* larvae can be exposed to various toxic phytochemicals while feeding on the hemolymph of caterpillar hosts (e.g., potential exposure of *C. congregata* to insecticidal nicotine when parasitizing *Manduca sexta* feeding on tobacco; of *C. rubecula*, *C. glomerata* and *C. vestalis* to glucosinolates by developing in hosts consuming crucifers; and of *C. flavipes* to phenylpropanoids and flavonoids as parasitizing hosts on sugarcane). This surprisingly low diversification of the detoxification arsenal could suggest that wasp larvae may not be as exposed to toxic compounds as expected due to direct excretion of these chemicals by the host larvae[39,40] or to the sequestration of toxic compounds[41] in tissues not consumed by parasitoid larvae. It is also possible that some bracovirus virulence genes of unknown function might contribute to protect parasitoid larvae against toxic compounds.

*Cotesia* wasps face different immune challenges during their lifetimes. While feeding on nectar, the adult might be exposed to similar environmental challenges to honey bees. Development inside the caterpillar host could on the one hand shield wasp larvae from pathogens, but on the other hand expose them to opportunistic infections, because parasitism alters caterpillar immune defenses. Lastly, metamorphosis coincides with the production of bracovirus particles, against which wasp antiviral responses had so far not been investigated. Insects were recently shown to recognize their obligate bacterial symbionts as foreign and to exert strong immune control, as documented for *Sitophilus oryzae* Primary Endosymbiont (SOPE)[42]. As the immunity gene arsenal of *C. congregata* is comparable to that of the honey bee, this wasp is most probably able to mount an immune response against pathogens, including viruses. However, the transcriptomic approach did not reveal any significant difference in immune gene expression between the ovaries of different pupal stages, although massive amounts of bracovirus particles are produced from day 5. This might reflect a lack of ovary cells ability to react, or that virus particles are recognized as self. Whatever the

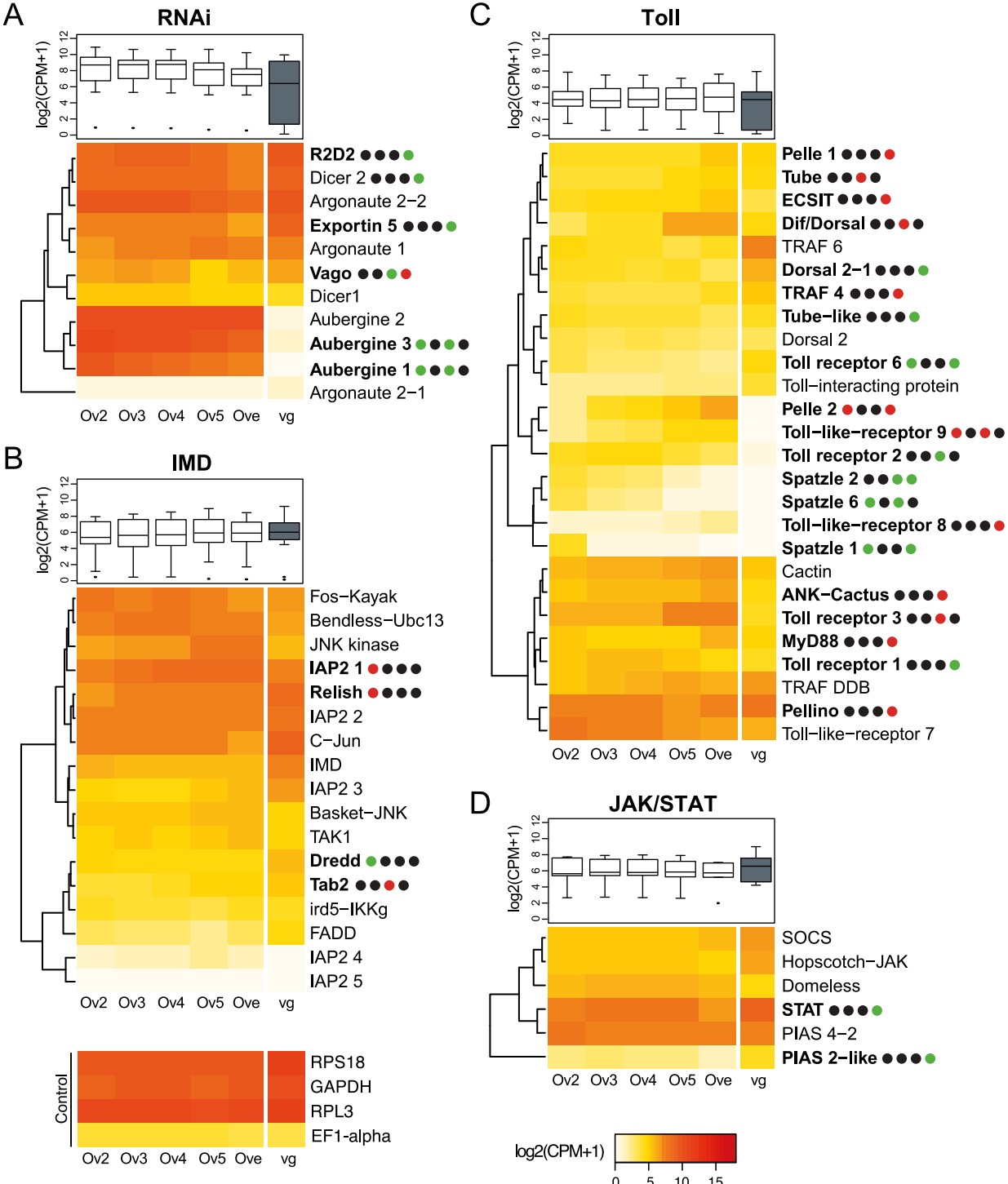

**Fig. 6 Gene expression of antiviral immune genes in the ovaries during *C. congregata* nymphal development.** The heatmaps show the expression of genes involved in **A** RNAi, **B** Imd, **C** Toll and **D** Jak-STAT pathways across the developmental stages of ovaries (Ov2, Ov3, Ov4, Ov5, Ove) and in venom glands (vg). The trees on the left are unsupervised hierarchical clustering of expression values. Boxplots represent overall expression of each pathway in ovaries and venom glands. Bold names highlight the genes that are differentially expressed between two stages and dots represent the four different comparisons studied between consecutive ovary stages (Ov2 vs. Ov3, Ov3 vs. Ov4, Ov4 vs. Ov5 and Ov5 vs. Ove). Black, red, and green dots indicate similar, increased and reduced expressions between consecutive developmental stages, respectively. Note that no particular trend appears correlated to bracovirus particles production, which occurs massively from Ov5 onward. The expression of four stable wasp genes having high (RPS18, RPL3, GAPDH) or low expression level (EF1-alpha) is presented as control for comparison.

mechanism involved, there is apparently no conflict remaining between the wasp and the virus after this ancient endogenization. We cannot exclude the possibility that immune cells from the hemolymph or fat body could perceive virus particle production and mount an immune response. However, this seems unlikely, since virus producing cells are tightly isolated by an epithelial layer and the ovary sheath: particles have not been observed in other wasp tissues and are not present in the hemolymph, they are exclusively released in the ovary lumen.

With the ancestral integration of a nudivirus genome, the parasitoid wasp gained a series of viral functions: including viral transcription, viral DNA replication, particle morphogenesis and infectivity. Whereas the function of viral DNA replication has been lost[5], thus impeding autonomous virus re-emergence, the other viral functions have been reorganized for virulence gene transfer via bracovirus particles. Chromosomal scale resolution of the *C. congregata* genome showed that bracovirus genes have colonized all the chromosomes with a nearly 2.5-fold increase in the total number of virus genes (nudiviral plus virulence genes) compared to the genome of a pathogenic nudivirus. This contrasts sharply with the decay of most viruses integrated into eukaryote genomes that do not provide a benefit to their host. Bracovirus dispersion occurred between 100 and 53 mya, as 25 viral loci are orthologous between *Cotesia* and *Microplitis* (Fig. 3 and Supplementary Fig. 7) and most of the proviral segments were already present in their common ancestor. The genomic organization stasis, observed since then, is reminiscent of bacterial symbiont genomes, which underwent major rearrangements specifically during the initial steps of association[43]. Yet, the organization of many bracovirus genes in clusters suggests that strong evolutionary constraints maintain these genes together. In the case of the nudiviral cluster, which encodes major capsid components (VP39, 38 K) on chromosome 7, DNA amplification, as a single unit[31], might enable mass production of bracoviral particles that are injected into parasitized hosts and accordingly many nudiviral cluster genes are the highest or among the most expressed genes in the ovaries. This could explain the counter selection on the dispersion or loss of these clustered nudivirus genes since the separation of the *Cotesia* and *Microplitis* lineages. In the case of the particularly large macrolocus, which comprises 77% of the virulence genes in the *Cotesia* genome, clustering could facilitate the evolution of new virulence genes copies by duplication[20], and thereby wasp adaptation against host resistance or to new hosts[5,29]. This organization may also promote the transmission of bracovirus virulence genes as a block of co-evolved genes, as shown for supergenes involved in butterfly wing patterns[44] and ant social behavior[45]. More generally our study shows that Bracovirus nudiviral cluster and proviral loci belong to remarkable genomic structures, the architecture of which are as evolutionarily constrained as supergenes, ribosomal DNA regions, Major Histocompatibility Complex, and chorion genes clusters. The next challenge will be to determine whether proximal causes are also underlying this organization as for example by further dissecting bracovirus DNA replication mechanism and identifying the role of the conserved regulatory signals we found at all replication unit extremities.

Remarkably, despite their semi-dispersed locations in the wasp genome, nudiviral genes remain synchronically expressed and under stabilizing selection, thus enabling the production of infectious bracovirus particles. This striking conservation of the viral machinery highlights the paramount importance of the production of viral particles allowing virulence gene transfer to a host, in the evolutionary history of the wasp. Strikingly, the stability of bracovirus loci in the wasp genome over 53 million years is in sharp contrast with recently reported high mobility of endogenous Ichnoviruses (IVs), which evolved within

ichneumonid wasp genomes from a different virus ancestor[46]. This difference might reflect characteristic features of the originally integrated virus, such as an ability to transpose, that could impact the evolution of viral sequences within wasp genomes. As an alternative hypothesis, IVs, contrary to BVs, may have not reached the stage leading to stabilization of viral loci in the wasp genome; indeed it is not known whether a single ancient or several recent endogenization events of viruses from the same family occurred in ichneumonid wasps[46]. In addition to IVs and BVs, several independent events of nudivirus captures have led to the production of viral liposomes allowing the delivery of virulence proteins to the parasitized host instead of virulence genes[47–51]. Comparisons between high-quality genomes of a variety of parasitoid wasps convergently associated with different viruses would be essential to reveal whether the evolution of beneficial large DNA endogenous viruses follows universal rules or each time a unique trajectory.

## Materials and methods

**Sampling**. The *C. congregata* laboratory strain was reared on its natural host, the tobacco hornworm, *M. sexta* (Lepidoptera: Sphingidae) fed using artificial diet containing nicotine as previously described[20]. *C. sesamiae* isofemale line came from individuals collected in the field in Kenya (near the city of Kitale) and was maintained on *Sesamia nonagrioides*[52]. *C. flavipes* individuals originated from the strain used for biological control against *Diatraea saccharalis* in Brazil[53]. *C. glomerata*, *C. rubecula* and *C. vestalis* laboratory cultures were established from individuals collected in the vicinity of Wageningen in Netherlands, and reared on *Pieris brassicae*, *Pieris rapae*, and *Plutella xylostella* larvae, respectively[54,55]. To reduce the genetic diversity of the samples prior to genome sequencing, a limited number of wasps were pooled; for example, only haploid males from a single virgin female were used for Illumina sequencing of *C. congregata* genome, ten female and male pupae originating from a single parent couple were used to generate *C. glomerata* genome, five male pupae originating from a single *C. rubecula* virgin female for *C. rubecula* genome and 40 adult males and 8 adult females from multiple generations of *C. vestalis* cultured in the Netherlands were used. DNAs were extracted from adult wasps and stored in extraction buffer following two protocols. *C. congregata*, *C. sesamiae* and *C. flavipes* DNA were extracted using a Qiamp DNA extraction kit (Qiagen) with RNAse treatment following the manufacturer's instructions and eluted in 200 μl of molecular biology grade water (5PRIME). *C. glomerata*, *C. rubecula* and *C. vestalis* DNA was extracted using phenol-chloroform.

**Genome sequencing and assembly**. *Cotesia congregata* genome was sequenced combining two approaches: (i) single-end reads and MatePair libraries of 3 Kb, 8 Kb, and 20 Kb fragments on 454 GS FLX Titanium platform (Roche) and (ii) paired-end reads of 320 bp fragments with HiSeq2000 platform (Illumina). *C. glomerata*, *C. rubecula* and *C. vestalis* DNA libraries were prepared using insert sizes of 300 and 700 bp. For *C. sesamiae* and *C. flavipes* libraries an insert size of 400 bp was selected. These libraries were sequenced in 100 bp paired-end reads on a HiSeq2000 platform (Illumina) at the French National Sequencing Institute (CEA/Genoscope, France) and at the Sequencing Facility of the University Medical Center (Groningen, Netherlands). Reads were then filtered according to different criteria: low-quality bases ($Q < 20$) at the read extremities, bases after the second N found in a read, read shorter than 30 bp and reads matching with phiX (Illumina intern control) were removed using in-house software (http://www.genoscope.cns.fr/fastxtend) based on the FASTX-Toolkit (http://hannonlab.cshl.edu/fastx_toolkit) as described in ref. [56].

The *C. congregata* genome was generated by de novo assembly of 454 reads using GS De Novo Assembler from the Newbler software package v2.8[57]. The consensus was polished using the Illumina data as previously described[58]. Gaps were filled using GapCloser module from the SOAPdenovo assembler[59]. The genomes of *C. sesamiae*, *C. flavipes*, *C. glomerata*, *C. rubecula* and *C. vestalis* were assembled with Velvet v1.2.07[57] using the following parameters: velveth *k*-mer 91 -shortPaired -fastq -separate, velvetg -clean yes and specific adapted values for -exp_cov and -cov_cutoff for each species.

### Chromosome scale assembly of *C. congregata* genome

*Hi-C library preparation*. The individual wasps had their gut removed and were immediately suspended after sampling in 30 mL of 1X tris-EDTA buffer and formaldehyde at 3% concentration, then fixed for 1 h. Ten milliliters of glycine at 2.5 M was added to the mix for quenching during 20 min. A centrifugation recovered the resulting pellet for −80 °C storage and awaiting further use. The libraries were then prepared and sequenced (2 × 75 bp, paired-end Illumina NextSeq with the first ten bases acting as tags), as previously described[60] using the *Dpn*II enzyme.

*Read processing and Hi-C map generation.* The Hi-C read library was processed and mapped onto *Dpn*II fragments of the reference assembly using HiCbox (available at https://github.com/rkoszul/HiC-Box) with bowtie2[61] on the back-end (option --very-sensitive-local, discarding alignments with mapping quality below 30). Fragments were then filtered according to size and coverage distribution, discarding sequences shorter than 50 bp or below one standard deviation away from the mean coverage. Both trimmed contact maps were then recursively sum-pooled four times by groups of three, yielding bins of $3^4 = 81$ fragments.

*Contact-based re-assembly.* The genome was reassembled using an updated version of GRAAL (dubbed instaGRAAL[62]) for large genomes on the aforementioned contact maps for 1000 cycles, as described[62]. Briefly, the program modifies the relative positions and/or orientations of sequences according to expected contacts given by a polymer model. These modifications take the form of a fixed set of operations (swapping, flipping, inserting, merging, etc.) on the 81-fragment bins. The likelihood of the model is maximized by sampling the parameters with a Markov Chain Monte Carlo (MCMC) method. After a number of iterations, the contact distribution converges and the global genome structure ceases to evolve, at which point the genome is considered reassembled. The process yielded eleven main scaffolds comprising >99% of the bin sequences.

*Polishing and post-assembly processing.* Each scaffold was independently polished by reconstructing the initial contig structure whenever relocations or inversions were found. In addition, previously filtered sequences were reintegrated next to their original neighbors in their initial contig when applicable. The implementation is part of instaGRAAL polishing and available at https://github.com/koszullab/instaGRAAL (run using the –polishing mode).

*Assembly validation.* We performed the validation with QUAST-LG[63], an updated version of the QUAST analyzer tailored for large genomes. The initial assembly from Illumina short reads was used as reference. The assessed metrics include genomic feature completeness, Nx and NGx statistics as well as global and local misassemblies. In addition, each assembly was assessed for ortholog completeness with BUSCO v3[64]. The reference dataset used for expected gene content was pulled from the OrthoDB (version 9) database for Hymenoptera, comprising 4,415 orthologs.

## Genome annotations

*Transposable element annotation.* Genome annotation was first done on the *C. congregata* reference genome and then exported on the genomes of the five other *Cotesia* species. First, the transposable element annotation was realized following the REPET pipeline comprising a de novo prediction (TEdenovo) and an annotation using TE libraries (TEannot)[65]. This annotation was exported into GFF3 files used as mask for the gene annotation.

*Automated gene annotation.* The automated gene prediction and annotation of *C. congregata* genome was done using Gmove (https://github.com/institut-de-genomique/Gmove) integrating different features based on (i) the mapping of Hymenoptera proteins from all hymenopteran genomes available on NCBI and UniProt Hymenoptera, (ii) the mapping of RNA-Seq data from *C. congregata*, *C. glomerata*, *C. vestalis*, and *C. rubecula* (this paper and PRJNA289655, PRJNA485865 PRJNA289731), and (iii) ab initio genes predictions using SNAP[66]. The automated annotation of the five other *Cotesia* species was performed using MAKER[67] using the same features as for the annotation of *C. congregata* but also including the output automated annotation of *C. congregata*.

*Automated gene functional annotation.* The functional annotation was performed using blastp from BLAST + v2.5.0[68] to compare the *C. congregata* proteins to the NCBI nonredundant database (from the 29/06/2014). The ten best hits below an e-value of 1e-08 without complexity masking were conserved. Interproscan v5.13-52.0[69] was used to analyze protein sequences seeking for known protein domains in the different databases available in Interproscan. Finally, we used Blast2GO[70] to associate a protein with a GO group (Supplementary Fig. 8).

*Specialist gene annotation.* The automated annotations were followed by manual curations, corrections and annotations realized by specialists of each gene family of interest through Apollo[71]. The genomes were available to this consortium through the web portal: https://bipaa.genouest.org/is/parwaspdb. Supplementary Data 2 summarizes the genes manually annotated by experts of different biological functions according to the phylogenetic level of interest for the comparisons. For the study of *Cotesia* immunity it was interesting to verify manually those of *C. congregata* genome to study whether they were induced after bracovirus particles production, the immune genes of other species were only automatically annotated.

*Genome completeness evaluation.* The completeness of the genomes and annotations were evaluated using Benchmarking Universal Single-Copy Orthologs BUSCO v3[64] using the insecta_odb9 database composed of 1658 genes. Contigs were searched for similarities against the nonredundant NCBI nucleotide (nt) (release November 2019) and the Uniref90 protein (release November 2019)

databases using, respectively, blastn from BLAST + v2.7.1[68] and diamond v0.9.29.130[72]. For both tasks, *e*-value cutoff was set to $10^{-25}$. Taxa were inferred according to the highest-scoring matches sum across all hits for each taxonomic rank in the two databases. Sequencing coverage was deduced after mapping Illumina paired reads to the assembly with Bowtie2 v2.3.4.2[61]. Contigs were then displayed with Blobtools v1.1.1[73] using taxon-annotated-GC-coverage plots.

**Orthologous genes identification, alignment, and phylogeny.** Orthologous genes between all genes annotated in the six *Cotesia* species and the four outgroups (*Microplitis demolitor*, *Nasonia vitripennis*, *Apis mellifera* and *Drosophila melanogaster*) were identified using OrthoFinder v1.14[74]. Universal single-copy ortholog genes from BUSCO v3[64] were extracted for the six *Cotesia* species and the four outgroups, aligned using MAFFT v7.017[75] and concatenated. The species phylogeny was performed on this alignment composed of 1058 orthologous for a length of 611 kb using PhyML program[76] with the HKY85 substitution model, previously determined by jModelTest v2.1[77] and the branch support were measured after 1000 bootstrap iterations. The cluster of orthologous genes was used to determine the phylogenetic level of each gene represented in Fig. 2. as follows: genes shared by all species are called shared by all; genes shared by at least nine species among the ten studied species without phylogenetic logical are named "shared by some"; genes shared by only Hymenoptera species and without orthologous gene in *D. melanogaster* are considered as "Hymenoptera specific"; genes shared only by Microgastrinae are named "Microgastrinae specific"; genes shared only by *Cotesia* species and without orthologous genes in any of the outgroup are considered as "Cotesia specific".

**Synteny analyses.** The different synteny analyses were performed on the orthologous genes identified by OrthoFinder v1.14[74] and by reciprocal blastp from BLAST + v2.2.28 on the annotated proteins (e-value below 10e$^{-20}$). The correspondence between the genes, the localizations on the scaffold and the figures were realized thanks to a custom R script (R Core Team 2013).

**Evolution of gene families.** For OR, P450 and *odv-e66* genes manually annotated genes from the reference genome of *C. congregata* were used along with orthologous genes from the five other *Cotesia* species, *M. demolitor*[78], *N. vitripennis*[79], *A. mellifera*[80] to create a phylogeny of each family among Hymenoptera. Protein sequences were first aligned with MAFFT v7.017[75] and the maximum-likelihood phylogeny was performed with PhyML[76] using the JTT + G + F substitution model for OR and using HKY80 substitution model for P450 and *odv-e66* genes. The branch support was assessed using aLRT for OR and 1000 bootstraps for P450 and *odv-e66* genes. The trees were then rooted to Orco (OR-coreceptor) clade for OR and the midpoint for the other. The gene gains and losses along the phylogeny for the different gene families of interest were identified with NOTUNG v2.9[81] as described[82].

**Evolution of single-copy genes.** To determine evolutionary rates within *Cotesia* genus, single-copy orthologous gene clusters (BUSCO, nudiviral and virulence genes) were first aligned using MACSE[83] to produce reliable codon-based alignments. From these codon alignments, pairwise $dN/dS$ values were estimated between *C. congregata* and *C. sesamiae*, the two most diverging species in the *Cotesia* phylogeny, with PAML v4.8[84] using the YN00 program. $dN/dS$ of the different gene categories of interest were then compared using a Kruskal–Wallis test, and Nemenyi-Tests for multiple comparisons were realized with the R package. For the nudiviral genes the $dN/dS$ values were calculated using genes from the six species. Orthologous genes from the six *Cotesia* species were aligned as described before and codeml (implemented in PAML v4.8) was used to estimate the M0 $dN/dS$ (free ratio model). This model was compared to a neutral model for which the $dN/dS$ is fixed to 1.

## RNA-seq analyses

*Sample preparation, extraction, and sequencing.* The ovaries and venom glands were extracted from females at five pupal stages, i.e., days 2, days 3, days 4, days 5 and at emergence, corresponding to the number of days after the creation of the cocoon and identified following body melanization[36]. Ovaries were pooled by groups of 20 pairs and venom glands by 100 and in duplicates for each condition. Samples were stored in buffer provided in the extraction kit by adding β-mercaptoethanol to reduce RNA degradation. Extractions were performed using QIAGEN RNeasy kit following manufacturer's recommendations. RNA-Seq library preparations were carried out from 1 to 2 µg total RNA using the TruSeq Stranded mRNA sample prep kit (Illumina, San Diego, CA, USA), which allows mRNA strand orientation (sequence reads occur in the same orientation as anti-sense RNA). Briefly, poly(A) + RNA was selected with oligo(dT) beads, chemically fragmented and converted into single-stranded complementary DNA (cDNA) using random hexamer priming. Then, the second strand was generated to create double-stranded cDNA. cDNA were then 3'-adenylated, and Illumina adapters were added. Ligation products were PCR-amplified. Ready-to-sequence Illumina libraries were then quantified by qPCR using the KAPA Library Quantification Kit for Illumina Libraries (KapaBiosystems, Wilmington, MA, USA), and libraries profiles evaluated with an Agilent 2100 Bioanalyzer (Agilent Technologies, Santa

Clara, CA, USA). Each library was sequenced using 101 bp paired-end reads chemistry on a HiSeq2000 Illumina sequencer.

*Analyses*. The pair-end reads from *C. congregata* ovary and venom gland libraries were mapped on the reference genome using TopHat2[85] with default parameters. Then, featureCounts program from the Subread package[86] was used to determine fragment counts per genes using default parameters.

To analyze gene expression, the raw fragment counts of ovaries and venom glands samples were first converted to counts per million (CPM) using the edgeR-implemented package[87] (R-Core Team 2017). Expressed genes were filtered based on a CPM > 0.4 (corresponding to raw count of 15) in at least two of the libraries incorporated in the analysis (Supplementary Fig. 9A) and subsequent normalization was performed on CPMs using the edgeR TMM method for Normalization Factor calculation[88] (Supplementary Fig. 9B). The reproducibility of replicates was then assessed by Spearman correlation of gene expression profiles based on filtered and normalized CPMs (Supplementary Fig. 9C).

To examine differential expression between ovary stages and with venom glands a quasi-likelihood-negative binomial generalized log-linear model was fitted to the data after estimation of the common dispersion using edgeR. Then, empirical Bayes quasi-likelihood F-tests were performed to identify differentially expressed (DE) genes under chosen contrasts[89]. Finally, F-test p-values were adjusted using false-discovery rate (FDR) method[90]. Genes were considered as DE whether FDR < 0.05 and fold-change (FC) of expressions between compared conditions was higher or equal to 1.5. Four contrasts were designed between the five successive ovary stages and a control contrast was tested between ovaries and venom glands at wasp emergence stage.

**Statistics and reproducibility**. To obtain high-quality genome assemblies it is necessary to limit the variability of the samples used for sequencing, which was done as much as possible for the six *Cotesia* genomes reported, as described in the sampling section. Access to the variability of these genomes will require rese-quencing approaches. Different statistical analyses are deeply involved in nearly all steps of the study from genome assemblies to transcriptome analyses and reported in the corresponding sections.

**Reporting summary**. Further information on research design is available in the Nature Research Reporting Summary linked to this article.

## Data availability
The datasets and genomes generated during the current study are available from the European Bioinformatic Institute (EMBL-EBI) and National Center for Biotechnology Information (NCBI) at the following BioProject IDs: PRJEB36310 and PRJEB43234 (umbrella project PRJEB40240). Genome database (genomes and annotated genes) is also available on the web site BIPAA (Bioinformatic Platform for Agrosystem Arthropods) https://bipaa.genouest.org/is/parwaspdb/.

## Code availability
Custom scripts are available from https://github.com/JeremyLGauthier/Scripts_Cotesia_Genomes or on Zenodo doi: 10.5281/zenodo.4116412[91].

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

## Acknowledgements

We thank Paul André Catalayud for providing the pictures of C. sesamiae and C. flavipes Germain Chevignon for the pictures of C. congregata nymphal stages and Juline Herbinière for TEM images of C. congregata ovaries. We thank the ADALEP (Adaptation of Lepidoptera) network for the involvement of its members and access to bioinformatic facilities for genome annotation. C. sesamiae individuals used in this study originated from the icipe, under the juridical framework of a Material Transfer Agreement signed between IRD, icipe and CNRS (CNRS 072057/IRD 302227/00). The authors thank Bruno Le Ru, Gerphas Ogola, and Julius Obonyo, which made the insects accessible for the study. C. congregata, C. sesamiae, C. flavipes genomes sequencing were funded by French National Research Agency ANR (ABC Papogen project ANR-12-ADAP-0001 and CoteBio ANR17-CE32-0015-02 to L. Kaiser). C. rubecula, C. glomerata, C. vestalis

genomes sequencing were funded by NWO EcoGenomics grant 844.10.002 to L.E.M. Vet, NWO VENI grant 863.07.010 and Enabling Technology Platform Hotel grant to L.E.M. Vet. HiC approach was funded by ERC project 260822 to R. Koszul. *C. congregata* transcriptomic analysis was funded by APEGE project (CNRS-INEE) to J.-M. Drezen. J. Gauthier thesis was funded by ANR and Region Center-Val de Loire. Collaboration between French and Netherland laboratories was funded by French ministry of foreign affairs and "Nuffic" ("VanGogh" Project to J.-M. Drezen and L.E.M. Vet).

## Author contributions

DNA/RNA preparation: J.G., C.C.-D., K.L., H.B., F.C., B.L.M.; Sequencing assembly and automatic annotation: B.N., J.-M.A., V.B., M.A.C., J.G., A.B., F.L.; *C. glomerata*, *C. rubecula*, *C. vestalis* sequencing and first assembly: J.V.V., H.S., J.d.B., S.W., L.V.; HiC approach: L.B., M.M., R.K.; TEs annotation: A.H.-V. (group leader), J.A., I.L.; Bracovirus annotation: A.B. (group leader), J.G., P.G., K.M., T.J., D.B., C.B., S.M., E.A.H.; Bracovirus gene evolution and synteny analyses: J.G.; H.B.; Immune genes annotation: E.H. (group leader), H.B., G.D.; B.D., N.K.; Conserved bracovirus regulatory sequences: G.P.; Chemosensory genes annotation: E.J.-J. (group leader), M.H., E.P., N.M., M.S.-A., I.B., M.S., M.M., T.C.; Detoxification genes annotation: G.l.G. (group leader), F.H., D.S.; Web annotation online platform: A.B.; F.L.; RNA-Seq analyses: H.B., J.G.; Wasp phylogenetic and biological background: J.B.W.; Project writing for funding: S.D., C.L.K.-A., E.H., J.-M.D., H.S., L.V.; Project coordination: J.G. and J.-M.D.; Manuscript writing: J.G., E.H., J.-M.D. with contributions from all authors.

## Competing interests

The authors declare no competing interests.

## Additional information

[1]Institut de Recherche sur la Biologie de l'Insecte, UMR 7261 CNRS-Université de Tours, Faculté des Sciences et Techniques, Parc de Grandmont, 37200 Tours, France. [2]Geneva Natural History Museum, 1208 Geneva, Switzerland. [3]EAWAG, Swiss Federal Institute of Aquatic Science and Technology, Dübendorf, Switzerland. [4]Department of Terrestrial Ecology, Netherlands Institute of Ecology (NIOO-KNAW), Droevendaalsesteeg 10, 6708 PB Wageningen, The Netherlands. [5]Institut Pasteur, Unité Régulation Spatiale des Génomes, UMR 3525, CNRS, Paris 75015, France. [6]Sorbonne Université, Collège Doctoral, 75005 Paris, France. [7]Sorbonne Université, INRAE, CNRS, IRD, UPEC, Univ. de Paris, Institute of Ecology and Environmental Science of Paris (iEES-Paris), 75005 Paris, France. [8]Génomique Métabolique, Genoscope, Institut François Jacob, CEA, CNRS, Univ Evry, Université Paris-Saclay, 91057 Evry, France. [9]IGEPP, INRAE, Institut Agro, Univ Rennes, 35000 Rennes, France. [10]Univ Rennes, Inria, CNRS, IRISA, 35000 Rennes, France. [11]Applied Bioinformatics, Wageningen University & Research, Wageningen, The Netherlands. [12]Université Montpellier, INRAE, DGIMI, 34095 Montpellier, France. [13]Laboratoire de Biométrie et Biologie Evolutive Université de Lyon, Université Claude Bernard Lyon 1, CNRS, UMR 5558, 43 bd du 11 novembre 1918, bat. G. Mendel, 69622 Villeurbanne Cedex, France. [14]Université Paris-Saclay, CNRS, IRD, UMR Évolution, Génomes, Comportement et Écologie, 91198 Gif-sur-Yvette, France. [15]Université Côte d'Azur, INRAE, CNRS, ISA, 06903 Sophia-Antipolis, France. [16]Université Paris-Saclay, INRAE, URGI, 78026 Versailles, France. [17]Insect Interactions Laboratory, Department of Entomology and Acarology, Luiz de Queiroz College of Agriculture (ESALQ), University of São Paulo, Piracicaba, São Paulo 13418-900, Brazil. [18]Laboratory of Entomology, Wageningen University, P.O. Box 16, Droevendaalsesteeg 1, 6708 PB Wageningen, The Netherlands. [19]Evolutionary Genetics, University of Groningen, Nijenborgh 4, 9747 AG Groningen, The Netherlands. [20]Department of Entomology, 320 Morrill Hall, 505 South Goodwin Avenue, University of Illinois, Urbana, IL 61801, USA. ✉email: drezen@univ-tours.fr

