## [Peer Review File · Communications Biology]

Reviewers' Comments:

Reviewer #1:

Remarks to the Author:

This is a nice contribution to some recent papers on endogenous viruses in braconid wasps. The combination of a de novo assembly along with comparative genomics to related wasps with illumina-only assemblies provides a nice context to this species. Additionally, data using RNA-seq of venom and ovary tissue is well done. Overall, the annotation of several important gene groups, chromosome scale super-scaffolding, along with comparative genomics make this manuscript of sufficient quality for publication in Communications Biology.

A few comments:

The word "expert annotation" is used several times, i would replace with manual annotation, my assumption is that these gene groups were manual reviewed vs relying on automated annotation methods alone.

No where that i can find is the contig N50 or size reported. I may have missed this, but it is important so that the gene space and "gappiness" of the assembly can be assessed. The mean scaffold size for the velvet-only assemblies is presented, but my understanding is that these would only be contig assemblies, not scaffolded (table S1). please clarify this. The N50 of these assemblies is very poor, but as to be expected from illumina only data with no mated-paired libraries.

In figure S1, can you describe what the values are on the X and Y axis, they don't appear to be BP values and I didn't see a label. Aligning the chromosomes to the axis makes a suggestion that these values are bp or relative to the genome length.

Reviewer #2:

Remarks to the Author:

Gauthier and colleagues sequenced the genomes of six Cotesia species, and successfully generated an assembly at the chromosomal scale for Cotesia congregata. Their data shows that bracovirus genes colonized in all ten chromosomes which is the most interesting part of their study. They also examined viral and host gene expression profile and found no changes in the host immune related genes in response to their endosymbiotic partners.

I found this manuscript very interesting and well written. They described all essential information and the methodology and experimental design is very well established. I would recommend the current format of manuscript for publication.

Reviewer #3:

Remarks to the Author:

Brief summary of the manuscript and overall impression of the work

Gauthier et al in "Chromosomal scale assembly reveals symbiotic virus colonization of parasitic wasp genome" present an important study that improve greatly our knowledge regarding the genomic evolution of the fantastic biological model involving Braconid wasps and their Polydnavirus. They present the first chromosomal scale genome of a braconid wasp highlighting bracovirus gene colonization among all ten wasp chromosomes. They give insight into the evolution of two important

gene families involve in odorant receptor and in detoxification. They also analyzed for the first time the complete expression profile of the bracoviral genes during the time frame of viral particle production. Finally, they assessed the consequences of this massive virus production on the immune response of the wasp ovaries. This work is well constructed and well written and I have only few minor comments.

Specific comments

Results:

(1) I understand that this is not very useful but in general genomic papers present a pie chart or any plot fig with all the predicted genes classified by their potential function ... this would be valuable for the paper.

(2) In order to better replace expression level of the studied genes (bracovirus and immunity) among the expression level of the whole transcriptome it would be very valuable to highlight in each panel the expression level of a gene with the highest level of expression among each tissues / sample and also add a group of gene known to be very stable among tissues / sample (like housekeeping genes). This will allow authors to normalize the gradient colors among all panel and fig5/6. This info is clearly stated in the text (lines 273-277) but it is impossible to see it easily from the figs5/6 or from the table S3. Authors may want to color code table S3 for expression value similarly than in fig5/6.

(3) The RNAseq libraries preparation and sequencing performed by the Genoscope need more detail regarding the treatment of the total RNA obtained after RNA extraction (ie PolyA enrichment or Ribodepletion) and the procedure for cDNA construction (ie polyA based primer or random). Because the procedure could have important implication on the way the data are reported (ie Ribodepletion with random primer for cDNA synthetization means that an important variety of RNA could have been analyzed in the present study (ncRNA, miRNA, lncRNA)).

(4) A global statistic summary of the RNAseq data is missing. Number of raw reads per sample / number of reads mapped per sample with % / number of reads mapped in gene coding region with % / number of reads mapped in virus genes (nudivirus and segment) and may be the number of reads mapped in the gene families focused on in the study (Immunity ...) with % etc Then the authors will be able to discuss the effect of viral expression on the whole transcriptome and to assess the proportion of the transcriptome dedicated for virus production. The effect is not the same if 60% of the RNAseq reads map to viral genes or if reversely only 0.1% map to viral genes.

Discussion:

(5) Lines 347-351: In order to definitely conclude it would be valuable to study immune response expression in tissues known to be involved in the immune response (hemocyte and fatbody). Because the ovaries are not known to be involve in immune response ... I invite authors to add a sentence regarding this point here.

Figures:

(6) Fig1: In order to facilitate lecture of the panel A and the legend I suggest the author to highlight the different step of the description with numbers on the figure and in the legend.

(7) Fig3: Panel C and D I would choose a different color for odv-e66 (maybe orange) because hatched red with black bar is quite difficult to differentiate from red alone. Also, C and D characters are not

well aligned. I found somewhat difficult to follow the numbering of the different cluster between the different panel (1 / 2 / 3 nudiviral cluster / 4 / 7 / PL9 / PL6 / PL10 etc) this could be more efficient.

(8) Fig4: There is a problem with C and D citation in the legend.

(9) Fig5 and Fig6 : Again, in order to replace the expression level of the gene focused on those 2 figs I would normalize colors over the two fig (at least) or even over the whole transcriptome (better) to have an idea of the expression level of these genes compare to genes with the highest level of expression from the whole transcriptome.

(10) Fig 5: Panel C: names of the 2 replicates are missing at the bottom of the panel.

(11) Fig 6: It's look like there is two titles for this fig: "Gene expression of immune genes in the ovaries during *C. congregata* nymphal development" followed by "Gene expression of antiviral immunity genes during *C. congregata* development". Personally, I vote for the first one!

(12) Fig 6: The title of the panel A is "RNAi"? Instead of "Antiviral"?

(13) Table S2: Why none of the immunity genes were not detected in other Braconidae genomes? Any explanation of this would be useful somewhere.

(14) Table S3: Is this all the *C. congregata* genes or only those with expression in the studied tissues? It would be valuable to have the list of the genes with no expression in the studied tissues if any ...

Point by point reply to reviewers:

Reviewer #1 few comments:

The word "expert annotation" is used several times, i would replace with manual annotation, my assumption is that these gene groups were manual reviewed vs relying on automated annotation methods alone.

-Expert annotation has been replaced by manual annotation in the manuscript.

No where that i can find is the contig N50 or size reported. I may have missed this, but it is important so that the gene space and "gappiness" of the assembly can be assessed. The mean scaffold size for the velvet-only assemblies is presented, but my understanding is that these would only be contig assemblies, not scaffolded (table S1). please clarify this. The N50 of these assemblies is very poor, but as to be expected from illumina only data with no mated-paired libraries.

-There was a mistake in the legend. The numbers indicated are the "contig N50", "scaffold N50" is only available for *C. congregata* genome for which mated-paired libraries have been sequenced. The legend has been corrected accordingly in Table S1.

In figure S1, can you describe what the values are on the X and Y axis, they don't appear to be BP values and I didn't see a label. Aligning the chromosomes to the axis makes a suggestion that these values are bp or relative to the genome length.

-The axis values have been integrated in the figure S1 and its legend.

Reviewer #2) Specific comments

Results:

(1) I understand that this is not very useful but in general genomic papers present a pie chart or any plot fig with all the predicted genes classified by their potential function ... this would be valuable for the paper.

-Three pie charts summarizing main GO terms in the biological process, molecular function and cellular component categories (extracted from Blast2go results) have been included in the Supplementary Figures as fig. S8.

(2) In order to better replace expression level of the studied genes (bracovirus and immunity) among the expression level of the whole transcriptome it would be very valuable to highlight in each panel the expression level of a gene with the highest level of expression among each tissues / sample and also add a group of gene known to be very stable among tissues / sample (like housekeeping genes). This will allow authors to normalize the gradient colors among all panel and fig5/6. This info is clearly stated in the text (lines 273-277) but it is impossible to see it easily from the figs5/6 or from the table S3. Authors may want to color code table S3 for expression value similarly than in fig5/6.

-We have homogenized the colors in all panels and added a panel with four wasp genes (RPL3, RSP18, GAPDH and EF1-alpha) very stable among tissues/sample to better replace expression levels of the studied genes, both for figures 5 and 6. All heatmaps now have the same scale for expression levels. We had already two wasp genes in the nudiviral panel, their

products have been found within the particles of Chelonus inanitus bracovirus (CiBV) and we have more clearly stated that point by indicating their names as wasp genes instead of as particle components (27a = heat-shock protein beta-1 (HSP beta-1); 17b =nucleoside diphosphate kinase (NDK)).

(3)The RNAseq libraries preparation and sequencing performed by the Genoscope need more detail regarding the treatment of the total RNA obtained after RNA extraction (ie PolyA enrichment or Ribodepletion) and the procedure for cDNA construction (ie polyA based primer or random). Because the procedure could have important implication on the way the data are reported (ie Ribodepletion with random primer for cDNA synthetization means that an important variety of RNA could have been analyzed in the present study (ncRNA, miRNA, lncRNA ...).

-A paragraph has been added to the Material and Methods to provide information on libraries preparation for RNAseq analyses.

(4)A global statistic summary of the RNAseq data is missing. Number of raw reads per sample / number of reads mapped per sample with % / number of reads mapped in gene coding region with % / number of reads mapped in virus genes (nudivirus and segment) and may be the number of reads mapped in the gene families focused on in the study (Immunity ...) with % etc Then the authors will be able to discuss the effect of viral expression on the whole transcriptome and to assess the proportion of the transcriptome dedicated for virus production. The effect is not the same if 60% of the RNAseq reads map to viral genes or if reversely only 0.1% map to viral genes.

-A new figure, Fig. S6, has been added to the supplementary materials, it shows the numbers of reads and mapping rates for each RNAseq library as well as percentages of reads assigned to either nudiviral, virulence, immunity or other genes. As the wasp ovaries develop, we can see that the reads assigned to virus production (nudiviral category) increases from <1% to almost 15% of the total numbers of reads assigned to gene features.

Discussion:

(5)Lines 347-351: In order to definitely conclude it would be valuable to study immune response expression in tissues known to be involved in the immune response (hemocyte and fatbody). Because the ovaries are not known to be involve in immune response ... I invite authors to add a sentence regarding this point here.

-Particles are not released in the wasp body but are produced in the ovaries and exclusively released in the lumen, thus we did not really expect an immune response to be induced in hemocytes and fat body cells. However, since some epithelial cells have been described by TEM to display phagocytic properties, cleaning up cellular debris after lysis of virus particle producing cells, we tested whether a transcriptional response of immune genes could be observed in the ovaries. To better state this point and acknowledge reviewer criticism we have modified this point in the discussion by adding the following paragraph:

“Whatever the mechanism involved, there is apparently no conflict remaining between the wasp and the virus after this ancient endogenization. We cannot exclude that immune cells from the haemolymph or fat body could perceive virus particle production and mount an immune response. However, this seems unlikely as virus producing cells are tightly isolated by an epithelial layer and the ovary sheath. Furthermore, viral particles are exclusively released in the ovary lumen and have never been observed in other wasp tissues nor in the hemolymph.”

Figures:

(6) Fig1: In order to facilitate lecture of the panel A and the legend I suggest the author to highlight the different step of the description with numbers on the figure and in the legend.

-Numbers have been added to the figure and its legend.

(7) Fig3: Panel C and D I would choose a different color for odv-e66 (maybe orange) because hatched red with black bar is quite difficult to differentiate from red alone. Also, C and D characters are not well aligned. I found somewhat difficult to follow the numbering of the different cluster between the different panel (1 / 2 / 3 nudiviral cluster / 4 / 7 / PL9 / PL6 / PL10 etc) this could be more efficient.

-Odv-e66 is a specific gene family of nudiviral genes interesting for its particular diversification but they are still nudiviral genes, so we prefer to keep the red color in the Figures. However instead of the black bar we used small black dots, making Odv-e66 genes more easily identified among nudiviral genes. C and D have been aligned. Finally, the numbering has been simplified.

(8) Fig4: There is a problem with C and D citation in the legend.

-The missing legend describing part B has been added and the right order is thus restored.

(9) Fig5 and Fig6 : Again, in order to replace the expression level of the gene focused on those 2 figs I would normalize colors over the two fig (at least) or even over the whole transcriptome (better) to have an idea of the expression level of these genes compare to genes with the highest level of expression from the whole transcriptome.

-We have replaced the expression level of the genes of interest as suggested.

(10) Fig 5: Panel C: names of the 2 replicates are missing at the bottom of the panel. **-The replicate number, Ov2.1 and Ov2.2 have been added.**

(11) Fig 6: It's look like there is two titles for this fig: "Gene expression of immune genes in the ovaries during *C. congregata* nymphal development" followed by "Gene expression of antiviral immunity genes during *C. congregata* development". Personally, I vote for the first one!

-The second title has been removed.

(12) Fig 6: The title of the panel A is "RNAi"? Instead of "Antiviral"?

-The title has been changed to "RNAi".

(13) Table S2: Why none of the immunity genes were not detected in other Braconidae genomes? Any explanation of this would be useful somewhere.

-N.d. corresponds to "Not determined" genes. This table summarizes the genes manually annotated by experts of different functions. For the immune genes it was interesting to compare species with and without bracovirus and for this comparison the annotation of only one Braconidae species, i.e. *C. congregata*, was sufficient. But of course, these genes also exist in the other Cotesia species and can be retrieved in the automated functional annotation.

We have now clearly stated in the table S2 legend that manual annotation was performed mostly on *C. congregata*.

(14) Table S3: Is this all the *C. congregata* genes or only those with expression in the studied tissues? It would be valuable to have the list of the genes with no expression in the studied tissues if any ...

-Table S3 has been modified and now includes all the 14140 genes annotated in the *C. congregata* genome. The genes that were not expressed in any tissue are identified by “NE” for “not expressed” in the columns corresponding to expression levels. Since expression levels were too low to pass the filters in all analyzed tissues (expression level = 0) these genes were not included in expression analyses. In addition, a heatmap has been added according to the expression levels to facilitate visualization in the table.

Best Regards,

J-M D